# Orbital-flop transition of superfluid ³He in anisotropic silica aerogel

M. D. Nguyen [1]✉, Joshua Simon[1], J. W. Scott [1], A. M. Zimmerman [1], Y. C. Cincia Tsai[1] & W. P. Halperin [1]✉

Superfluid ³He is a paradigm for odd-parity Cooper pairing, ranging from neutron stars to uranium-based superconducting compounds. Recently it has been shown that ³He, imbibed in anisotropic silica aerogel with either positive or negative strain, preferentially selects either the chiral A-phase or the time-reversal-symmetric B-phase. This control over basic order parameter symmetry provides a useful model for understanding imperfect unconventional superconductors. For both phases, the orbital quantization axis is fixed by the direction of strain. Unexpectedly, at a specific temperature $T_x$, the orbital axis flops by 90°, but in reverse order for A and B-phases. Aided by diffusion limited cluster aggregation simulations of anisotropic aerogel and small angle X-ray measurements, we are able to classify these aerogels as either "planar" and "nematic" concluding that the orbital-flop is caused by competition between short and long range structures in these aerogels.

Disorder in quantum condensed states can be responsible for instability of thermodynamic phases[1], as has been predicted and shown to be the case in superfluid ³He in the presence of high porosity silica aerogel[2,3]. Similarly, disorder in cuprate superconductors in a magnetic field results in the vortex glass phase[4], and the role of macroscopic anisotropy from crystal twinning can influence the vortex structure[5]. We have shown that random isotropic disorder imposed on superfluid ³He imbibed in aerogels, destabilizes the chiral ³He A-phase in zero magnetic field[6,7]. Phase stability can be further tuned by introducing anisotropy, either by positive strain (stretching) which favors the chiral A-phase[8], Fig. 1a, or negative strain (compressing) which stabilizes the time-reveral symmetric isotropic B-phase[9], Fig. 1b. In another example of the strong influence of anisotropic impurity, the polar phase of superfluid ³He was discovered in the highly anisotropic $Al_2O_3$ aerogel (Nafen)[10] where half-quantum vortices have been identified[11] and predicted to harbor Majorana zero modes. In brief, pure superfluid ³He is a paradigm for odd-parity and unconventional BCS paired states and as such, superfluid ³He in aerogel can serve as a guide for understanding the effect of impurities and their anisotropy in superconductors.

The effect of aerogel structure on the superfluid has been described with quasiparticle scattering models[12,13], parametrized by the quasiparticle elastic mean free path and an impurity correlation length, also extended to include anisotropic scattering[14,15]. Experimentally, the structure has been investigated with small angle X-ray scattering (SAXS)[16,17], complemented by three-dimensional numerical simulations of the formation of the gel based on isotropic diffusion-limited cluster aggregation (DLCA) algorithms[18–22]. Here, we present a framework for simulating, analyzing, and classifying uniaxial anisotropic aerogels. In these simulated aerogels, we find a dominant large scale behavior that is consistent with SAXS data of lab grown aerogel[17]. We propose that this structure is responsible for the orbital analog of the spin-flop textural transition in superfluid ³He, denoted $T_x$[23], shown in Fig. 1c. We attribute the mechanism to the aerogel structure since it predicts a common behavior for both A and B-phases as observed in Fig. 1d.

Our DLCA simulations create an aerogel network by a process similar to that described by Hasmy et al.[20] which we summarize in the following section. To simulate anisotropic aerogel, we modify this procedure by biasing the diffusion of the constituent particles along one axis taken to be the z-axis. The degree of anisotropy is labeled by a single continuous variable, $\boldsymbol{\epsilon} = \epsilon \hat{\mathbf{z}}$, with $\epsilon$ defined to be the ratio of diffusivity along the z-axis to the diffusivity perpendicular to z. Isotropic aerogel has an anisotropy parameter of $\epsilon = 1$. Samples with $\epsilon > 1$

¹Department of Physics and Astronomy, Northwestern University, Evanston, IL 60208, USA. ✉e-mail: mannguyen2019@u.northwestern.edu; w-halperin@northwestern.edu

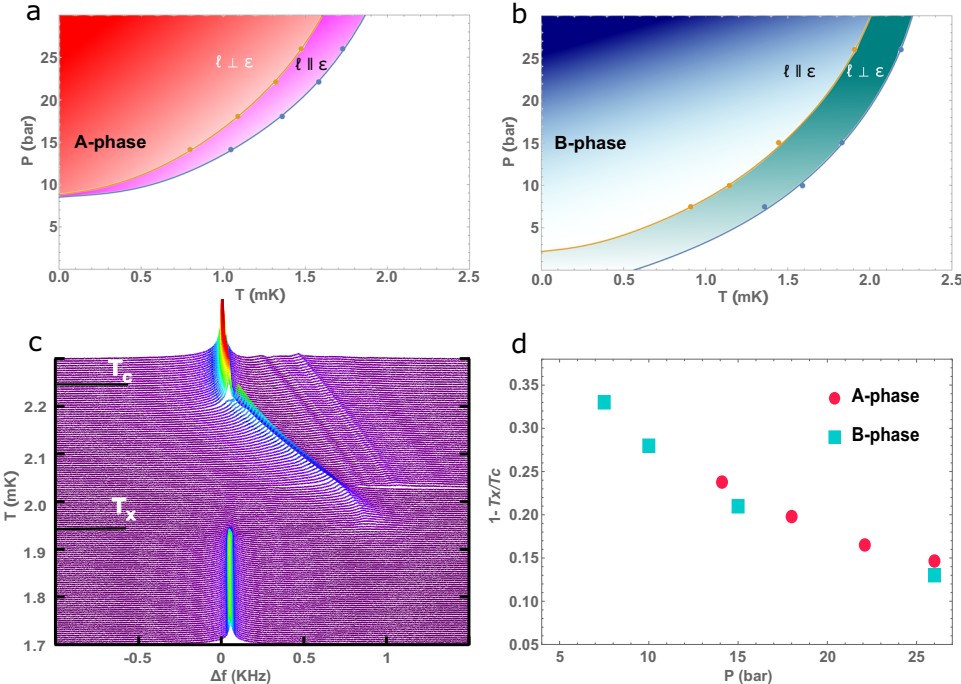

**Fig. 1 | Phase diagrams for superfluid ³He in 98% porous anisotropic silica aerogel in the limit of zero magnetic field. a** Pressure–temperature phase diagram for 14% positive strain which stabilizes the chiral A-phase[8]. Solid blue line and circles indicate $T_c$ while orange line and circles indicate $T_x$. The orbital angular momentum, $\ell$, swaps orientation above and below $T_x$. **b** Phase diagram similar to **a** for 20% negative strain which stabilizes the time-reversal symmetric B-phase[23].

The orientation of $\ell$ is reversed for this system. **c** Temperature dependence of the nuclear magnetic resonance (NMR) frequency spectrum of ³He, with ⁴He preplating to achieve specular surface conditions, as was used to determine the phase diagram of the B-phase[23], showing both transitions[23]. **d** The phase diagram for the flop transition temperature relative to the superfluid transition, $T_c$ without ⁴He, demonstrates universal behavior independent of order parameter symmetry.

and $\epsilon < 1$ have markedly different large scale structures representing two classes of anisotropic aerogel. These structures have distinct signatures in their correlation function and structure factor which can be directly calculated from the aerogel network. The structure factor is a particularly useful metric as it can be compared with small-angle X-ray scattering (SAXS) data from real amorphous materials like silica aerogel[20,24,25]. The SAXS measurements on anisotropic aerogels show clear anisotropy in the scattering pattern but the underlying structure can not be determined since scattering data is proportional to the amplitude of the scattered wave where all phase information is lost[26]. Consequently, it is not possible to fully reconstruct the underlying structure from SAXS data alone[27]. On the other hand, starting from simulated aerogel with a well-defined microscopic structure, we can calculate the structure factor and compare it with the SAXS measurements. We leverage this connection to classify silica aerogel as nematic or planar corresponding to the sign of the strain. The orientation of the order parameter of superfluid ³He in anisotropic silica aerogel depends importantly on this classification. We demonstrate that for either case, the structure induces the orbital-flop transition where the flop orientation depends on the sign of anisotropy imposed by strain, shown in Fig. 1a, b.

## Results

We detail the simulation procedure for these anisotropic aerogels in the Methods section. The resulting simulated cluster is a field denoted $\rho(\mathbf{r})$. For a discrete set of silica spheres,

$$\rho(\mathbf{r}) = \begin{cases} 1, & \frac{|\mathbf{r}-\mathbf{r_i}|}{\varrho_i} \leq 1 \, \forall \, i \in [1, ..., N] \\ 0, & otherwise \end{cases} \tag{1}$$

where $\varrho_i$ is the radius and $\mathbf{r_i}$ is center of the $i^{th}$-particle. This is stored numerically simply as a list of the $\mathbf{r_i}$ and $\varrho_i$. As these spheres aggregate, they form more complex structures which create a hierarchy of coarse

graining. At a microscopic level, the spheres tend to aggregate into quasi-one dimensional collections which we call strands. At the highest level for anisotropic aerogels, the strands are seen to cluster into nematic and planar structures which we detail below. In Fig. 2 we show a sample aerogel field for isotropic aerogel revealing the emergent order.

The full 3-D rendering of $\rho(\mathbf{r})$ obscures the strand and clustering of the aerogel network so 2-D, orthogonal projections are used to better visualize the spatial variation in the aerogel. The right-hand panel of Fig. 2 shows the highly-correlated distribution of particle position, complex strand structure, and characteristic voids in the aerogel network. For the case of isotropic aerogel, these properties have no preferred direction in space. In this work, we show that anisotropy can be introduced by biasing the diffusion step size; $\epsilon > 1$ indicates faster diffusion (larger step size) along the $\mathbf{z}$-axis while $\epsilon < 1$ indicates faster diffusion in the $xy$-plane.

For anisotropic aerogel, Fig. 3 reveals clear spatial anisotropy and large scale structure not found in the isotropic samples. We have numerically created two types of anisotropic aerogels with uniaxial, anisotropic diffusion which we classify as nematic, with $\epsilon < 1$, and planar, with $\epsilon > 1$. As seen in the the projected view of $\rho$ in Fig. 3, anisotropic diffusion introduces a preferred direction breaking the full 3-D large-scale rotational symmetry of isotropic aerogel. The aerogel strands are depicted as simple cylinders in the inset cartoon to help visualize the structure. A quantitative description of their properties is given in the following sections. In the case of $\epsilon < 1$, the strands are preferentially aligned along the anisotropy direction $\boldsymbol{\epsilon}$. This is akin to nematic liquid crystals where long molecules have orientational order along one axis and an absence of regular spatial ordering in the perpendicular plane.

For $\epsilon > 1$, the projected view along the $x$ and $y$-axes reveals high-density, planar sheets of aerogel clustered together with some characteristic thickness. These sheets are separated from each other by

**Fig. 2 | Real and simulated isotropic aerogel. a** Scanning electron microscopy of real, 98% porous isotropic aerogel shows the complex network of silica particles. **b** Simulated aerogel cluster for isotropic diffusion for a small segment of the sample (~0.5% of the total sample). Structural properties such as strand orientation, clustering, and void size are difficult to determine in the 3-D representation for the full sample. **c** Projecting the cluster onto orthogonal, 2-D planes reveals the position of silica spheres to be highly correlated. Each plane represents a projection of the aerogel sample along the axis perpendicular to that plane. For isotropic diffusion, the strands of silica appear to be without a preferred direction; however, characteristic clustering and void sizes are visibly apparent.

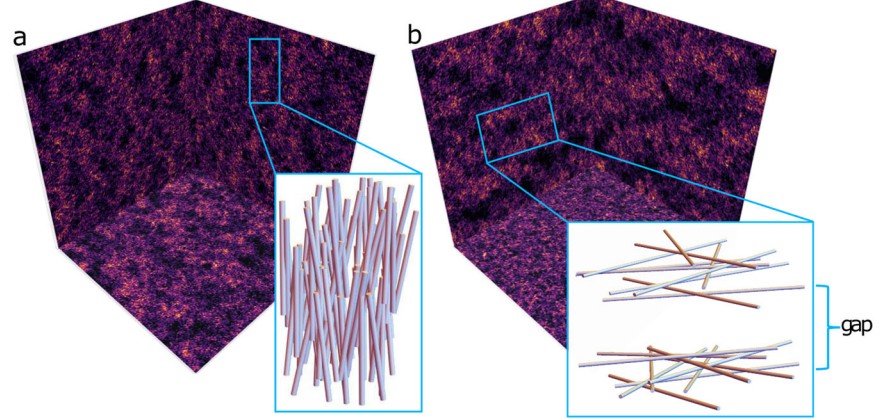

**Fig. 3 | Projection of aerogel structure for anisotropic aerogel.** The projections reveal clear anisotropic strand structures for $\epsilon \neq 1$ (as compared with isotropic aerogel in Fig. 1c). Illustrative cartoons in the insets depict the strands as cylinders. **a** For $\epsilon = 0.25$, the projections onto the $xz$- and $yz$-planes reveal long structures parallel to the anisotropy direction, $\boldsymbol{\epsilon}$, corresponding to nematic strands. The projection along $z$ into the $xy$-plane reveals strands oriented along $z$ correlated in their positions in the $xy$-plane. **b** For $\epsilon = 4$, sheets of aerogel strands form in the $xy$-plane, perpendicular to $\boldsymbol{\epsilon}$ with gaps between sheets (inset). The projection along $z$ into the $xy$-plane reveals a random structure indicating that the orientation of the strands between distantly separated sheets are uncorrelated.

visible gaps of low density regions with fewer particles. We classify samples with $\epsilon > 1$ as planar aerogels. This structure can be thought of as being analogous to a smectic liquid crystal. Quantitative measures of the spatial variation of these anisotropic aerogels that confirm this identification are given in the Methods Section.

Importantly, these two classifications of aerogel have been realized in the form of silica aerogels grown with 98% porosity in cylindrical form. A uniform uniaxial anisotropy can be imposed on an isotropic silica aerogel by compression, producing negative strain. Alternatively, growing the alcogel precursor of the aerogel using an excess of catalyst, leads to radial shrinkage during the final step of supercritical drying. This produces a stretched, or positively strained silica aerogel[17]. Furthermore, with compression, a stretched aerogel evolves smoothly to becoming isotropic and then, ultimately, a compressed aerogel with negative strain. This behavior was determined from SAXS and optical birefringence measurements[17].

## Structure factor

The structure factor, $S(\boldsymbol{q})$, establishes a connection between simulated and real aerogel. The X-ray scattering intensity, $I(\boldsymbol{q})$, can be decomposed as $I(\boldsymbol{q}) \propto S(\boldsymbol{q})F(\boldsymbol{q})$. $F(\boldsymbol{q})$ is the single-particle form-factor that encodes details about the particle shape which affects the large-$q$ behavior of $I(\boldsymbol{q})$. $S(\boldsymbol{q})$ encodes correlation in position of particles at intermediate and large spatial scale (small-$q$). For small-angle X-ray

scattering (SAXS) $S(\boldsymbol{q})$ dominates $I(\boldsymbol{q})$. Therefore, the structure factor can be used to directly compare with SAXS data.

There is extensive literature determining $S(\boldsymbol{q})$ by calculating the autocorrelation function, $g$, (see Methods section) and performing a Fourier transform[20,28,29]. This is usually done in spherical coordinates where the integration kernel simplifies from $e^{i\boldsymbol{q}\cdot\boldsymbol{R}}$ to $\sin(qr)/(qr)$ because the aerogel under consideration is isotropic. In the case of an anisotropic system, it is easier to numerically perform this Fourier transform in cartesian coordinates. This is done by first converting $\rho(\boldsymbol{r})$ into a sparse 3-D matrix, $\rho_{ijk}$, whose indices form a lattice and whose matrix elements represent the density at lattice site $(i, j, k)$. This matrix $\rho_{ijk}$ is numerically Fourier transformed into its conjugate field $\hat{f}_{lmn}$ with the cartesian $S_{xyz}$ given by[30]

$$S_{xyz} = |\mathcal{F}\{\rho_{xyz}\}|^2 = |\hat{f}_{lmn}|^2. \tag{2}$$

To compare with SAXS data for X-rays incident perpendicular to $\boldsymbol{\epsilon}$, $S_{xyz}$ is then converted to $S(\boldsymbol{q}_\parallel, \boldsymbol{q}_\perp)$, where $\boldsymbol{q}_\parallel$ is the component parallel to $\boldsymbol{\epsilon}$ and $\boldsymbol{q}_\perp$ is the perpendicular component.

The calculated structure factors of simulated aerogel in Fig. 4a–c are compared with the SAXS data from real aerogel Fig. 4d–f. Isotropic aerogel with $\epsilon = 1$ has the expected isotropic $S(\boldsymbol{q})$. For anisotropic samples, the structure factor has two features of note. First, the distinct dipolar angular distribution pattern at short $q$ (corresponding to a length scale of ~100 $r_0$) is evident in yellow and orange in (**b, c**).

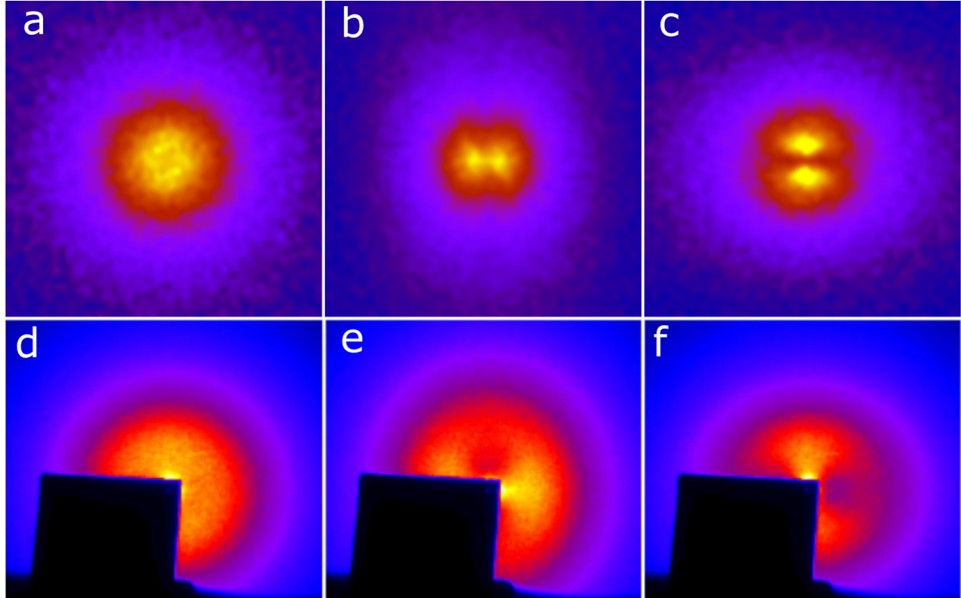

**Fig. 4 | Calculated $S(q)$ versus small angle X-ray scattering data[31].** Anisotropy axis is vertical. The top panels display the calculated structure factor $S(q)$. (**a**) isotropic; (**b**) nematic ($\epsilon = 0.25$), and (**c**) planar ($\epsilon = 4$) (axes and scale in Supplementary Materials). For the nematic simulation, at short $q$, $S(q)$ has a dipolar shape with the "long axis" perpendicular to $\epsilon$. For the planar case, at short $q$, $S(q)$ has a dipolar shape with the "long axis" along the anisotropy direction. However, at larger $q$, as seen in the purple regions, the "long axis" of the anisotropy pattern is rotated by 90°. The bottom panels display the small-angle X-ray scattering (SAXS) for lab-grown aerogel samples, adapted with permission from Elsevier: (**d**) isotropic, (**e**) axially compressed (12.7 % negative strain), and (**f**) stretched (13.7% positive strain) from ref. 17. The black square in the data images is the shadow of the X-ray beam stop, there is no data in those regions.

Secondly, the ellipsoidal pattern at intermediate $q$ (corresponding to ~10 $r_0$) can be seen in the purple regions. The orientation of these two patterns can be used to unambiguously classify aerogels. Nematic-like aerogels will have the short-$q$, dipolar pattern perpendicular to the anisotropy while planar-like aerogels will have the dipolar pattern parallel to $\epsilon$. This is a general framework for classifying aerogel and will be the case irrespective of the material or type of aerogel. The orientation of the anisotropy of the structure factor is a general feature encoding the difference between nematic and planar correlations. At larger $q$ (in the purple region of panels **b** and **c**), the major ("long") axis of the ellipsoidal pattern is rotated 90° from the short-$q$ dipole pattern. Evidently, there are two scales of structure for the aerogel. The large-scale structure is reflected in the dipole scattering pattern and the smaller scale structure oriented perpendicular to the large structure is reflected in the ellipsoidal pattern of the SAXS structure factor. We use the large-scale behavior to classify and label the aerogel samples as being either nematic or planar.

**Comparison with SAXS Data**

Figure 4d–f show the SAXS data for real aerogels. The two types of anisotropic aerogel analyzed are obtained by either compressing (negative strain) or stretching (positive strain) isotropic aerogels[31,32]. It was not previously known how this strain affected the underlying structure of aerogel. Comparing the SAXS data to our calculated $S(q)$, we can determine if compressing aerogel creates nematic or planar structures. Compressed aerogels, seen in (**e**), has the dipolar scattering pattern at short $q$ with the "long" dipole axis perpendicular to the anisotropy axis. For stretched aerogels in panel **f**, the short $q$ dipole pattern is parallel to $\epsilon$ while for intermediate $q$, the "long" axis is perpendicular to $\epsilon$. In other words, the direction in which scattering is more intense rotates by 90° as $q$ increases (going to smaller length scale), which is the same behavior observed in the calculated structure factors. The structure factor of simulated aerogel for q larger than displayed in Fig. 4 is shown in Supplementary Materials.

Comparing these SAXS patterns to the structure factor of simulated aerogels, we can identify the structure of experimentally produced aerogels. Stretched aerogel (positive strain) is consistent with planar aerogel. On the other hand, compressing isotropic aerogel unexpectedly leads to the formation of strands *along* the compression axis. This is contrary to Volovik's suggestion[2]. In his model, stretching aerogel would create long nematic strands parallel to the anisotropy axis while compressing aerogel would collapse the strands into planes. Here we find the opposite behavior. Our identification however is consistent with experimental results of superfluid ³He imbibed in anisotropic aerogels[8,9,23,33] which we discuss in the final section.

**Free Path Distribution**

The correlation function (see Methods) and structure factor characterize the aerogel structure itself. For many applications of aerogel, it is the void between the silica particles that is relevant rather than the aerogel network. An important measure of this negative space is the distribution of geometric free paths through the aerogel (which appears as a parameter in theoretical calculations of properties of ³He in aerogel[12,34]). That is to say, starting at the surface of a random particle, how far can a test ray move before colliding with the aerogel network? The condition for collision between a ray and a sphere is given by the discriminant[35]:

$$disc = (\hat{\boldsymbol{d}} \cdot (\boldsymbol{p}_f - \boldsymbol{p}_i))^2 - (|\boldsymbol{p}_f - \boldsymbol{p}_i|^2 - r_f^2) \geq 0 \qquad (3)$$

where $\boldsymbol{d} = d\hat{\boldsymbol{d}}$ is the ray, $\boldsymbol{p}_i$ is the origin of the ray, and $\boldsymbol{p}_f$ is the center of the final sphere with a radius $r_f$. If $disc \geq 0$ and $\hat{\boldsymbol{d}} \cdot (\boldsymbol{p}_f - \boldsymbol{p}_i) \geq 0$ (this second condition ensures that only collisions in the forward direction are considered), then the path length, $d$, is calculated as $d = (\hat{\boldsymbol{d}} \cdot (\boldsymbol{p}_f - \boldsymbol{p}_i)) - \sqrt{disc}$. The free path is determined by taking the minimum $d$ observed along the direction of travel. If no collision is observed within the initial box, periodic boundary conditions are applied. The bounding box plane that the ray intersects is determined, and then the aerogel sample is shifted in the appropriate direction and collision detection is applied for the shifted sample. This is repeated until a collision is found. A probability density function, $P(\boldsymbol{d})$, is obtained by taking a histogram of the catalog of free paths. A random

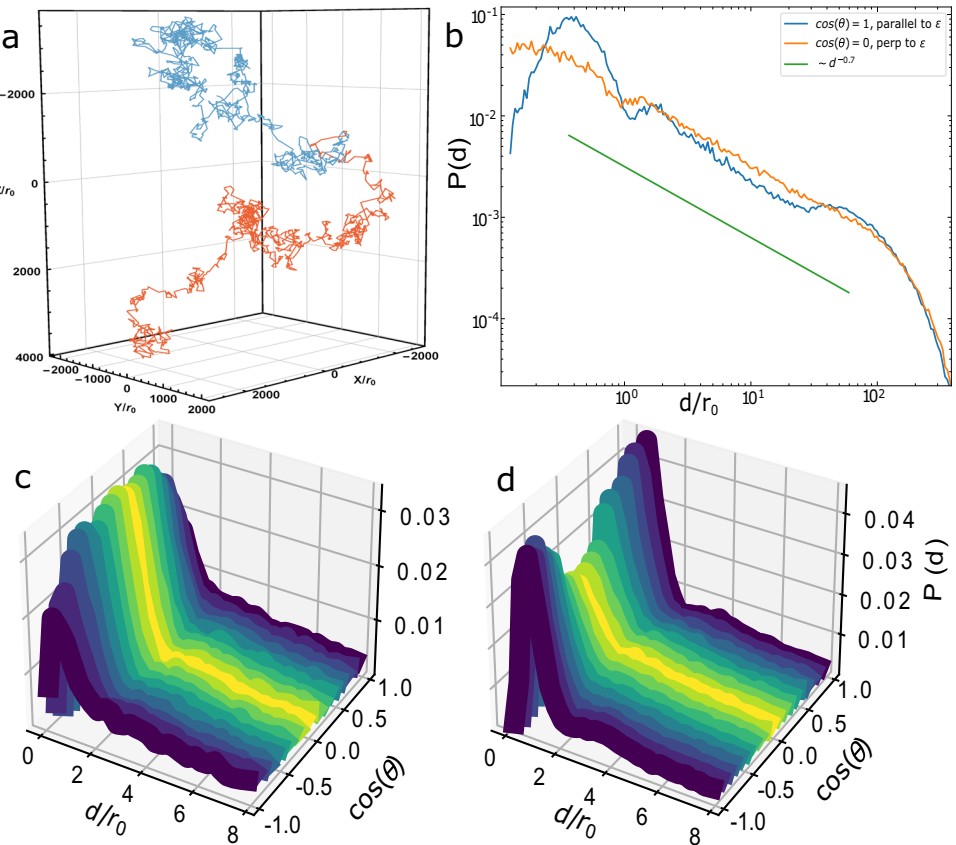

**Fig. 5 | Geometric free paths in the aerogel network. a** Examples of two free flights through nematic ($\epsilon = 0.25$) aerogel starting at the same location for different angles. **b** Distribution of free path for planar ($\epsilon = 8$) aerogel parallel (blue curve) and perpendicular (orange curve) to $\boldsymbol{\epsilon}$. There is a power-law path distribution ($\sim d^{-0.7}$)

for short paths before being exponentially cut off above $100\,r_0$. The bottom panels show the full $P(d, \theta)$ for ($\epsilon = 0.25$) (**c**) and ($\epsilon = 4$) (**d**) with a similar nearest-neighbor angular dependence on the correlation function, see Methods Section.

walk through the aerogel will have a distribution of step sizes given by $P(\boldsymbol{d})$. Figure 5a shows the diverging path of two random walks through nematic ($\epsilon = 0.25$) aerogel starting at the same particle but at different angles. These random walks exhibit the key feature of what are called "Lévy flights"[36,37]. The characteristic "jumps" of a Lévy flight are observed where the test ray is confined to small regions followed by big jumps to other regions[37].

From $P(\boldsymbol{d})$, we can calculate a mean free path, $\lambda$, which has been shown to be inversely proportional to density, for low density samples[38]. We find that while the distribution of free paths is very different for high porosity aerogel compared with a uniform Poisson point field of the same density, the mean free path for both systems are similar to within 7% of $60\,r_0$. There are two reasons for this. First, at low density, both a highly correlated system like aerogel and an uncorrelated uniform distribution will have large lines of sight. The excess correlation of the aerogel structure, which affects the distribution at short path lengths, has a smaller effect than density variations. Secondly, aerogel ceases to be a fractal above what is called the "upper-fractal cutoff"[28]. If aerogel had no upper fractal cutoff, then the free path distribution would be scale-free and described by a generalized Lévy distribution with the asymptotic form $P(d) \sim d^{-\alpha}$, with $1 < \alpha < 3$[36,37]. This power-law distribution is fat-tailed meaning that there is significant weight of the distribution in long free paths with the possibility that the mean is undefined (for $\alpha \leq 2$).

**Anisotropic free path**

However, both SAXS data and theoretical calculations show that aerogel is not fractal at all lengths[28,31]. Correspondingly the free path distribution is cut off and the mean is well defined. These "truncated

Lévy flights" however still retain many properties of Lévy flights such as super-diffusion[39]. As seen in Fig. 5, the distribution is indeed power-law below $100\,r_0$ with a very weak exponent of $\alpha = 0.7$ indicating a very flat probability distribution. At longer length scales above $\sim 100\,r_0$, the distribution is exponentially cut off. The cutoff is not due to finite size effects of the simulation as it remains constant with increasing $L/r_0$ from 100 to 350.

Uniaxially anisotropic aerogels, $P(\boldsymbol{d}) \Rightarrow P(d, \theta)$, are both functions of path length, $d$, and polar angle, $\theta$. The height of the peak of the distribution has a $\theta$-dependence similar to what is observed for the correlation function discussed below in the Methods Section. Also like the correlation function, the behavior of $P(d, \theta)$ for intermediate distances $\sim 20\,r_0$ is different from short distances. For planar aerogel, Fig. 5b, there are more free paths along $\boldsymbol{\epsilon}$ at very short distances but more free paths perpendicular to $\boldsymbol{\epsilon}$ at intermediate distances, consistent with the existence of large planar gaps in the structure as indicated in Fig. 3b (see also Methods Section).

Each angle can be considered an independent probability distribution and distribution moments can be defined at different angles. Two directions of particular interest are the mean free path parallel, $\lambda_\parallel$, and mean free path perpendicular, $\lambda_\perp$, to the anisotropy direction $\boldsymbol{\epsilon}$. Despite very clear anisotropy, the first moments of $P(d, \theta)$ (mean free path along a certain direction) are similar for each of the two orthogonal directions. For $\rho_0 \sim 2\%$, $\lambda_\parallel \sim \lambda_\perp \sim 60\,r_0$ in both the nematic and planar aerogels. For real silica aerogels used in superfluid $^3$He experiments, $r_0$ is $\sim 1.5 - 2$ nm, indicating a mean free path of $\sim 90-120$ nm[12]. In general, $r_0$ can be much larger, up to $\sim 10$ nm[40,41], consistent with experimental measurements for isotropic aerogel of comparable density[42]. Consequently, the mean free path is not a good parameter to

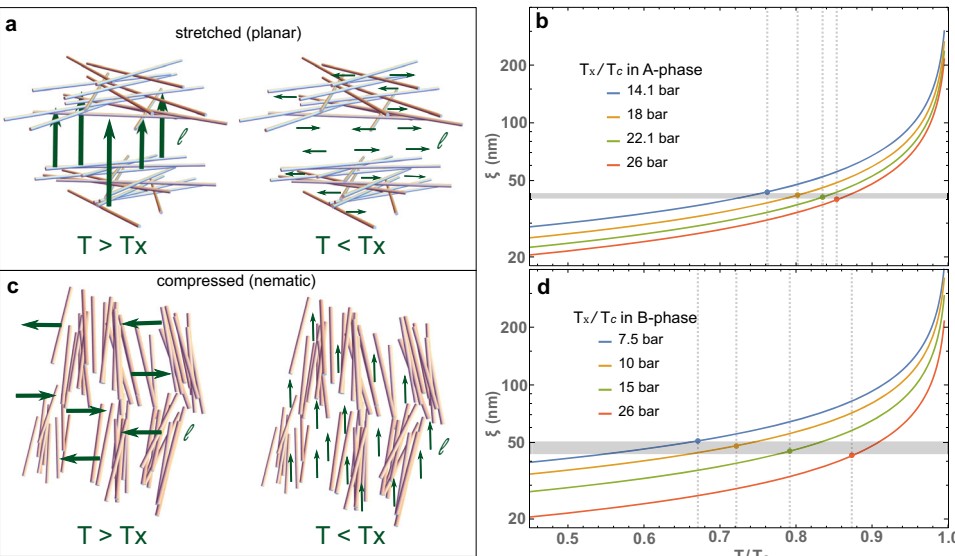

**Fig. 6 | Orbital angular momentum and coherence length. a** Orientation of the orbital angular momentum in stretched (planar) aerogel. Above $T_x$, the coherence length is large as suggested by the length of the green arrows for the orbital angular momentum $\boldsymbol{\ell}$ oriented perpendicular to the large scale planar structure. Below $T_x$, the coherence length is small and $\boldsymbol{\ell}$ reorients to being perpendicular to the small scale structure. **b** The Ginzburg–Landau coherence length $\xi(T, P)$ for various pressures. The orbital flop transition $T_x$ in the A-phase of stretched (planar) aerogel for each pressure is indicated by the data points and the vertical dashed lines. Although $T_x/T_c$ varies with pressure, the coherence length evaluated at various $T_x$, $\xi(T_x, P)$, all fall into a narrow band near 50 nm. This indicates that the orbital flop transition occurs when the superfluid coherence length decreases below that length scale. **c, d** Orientation of orbital axis and coherence length for the B-phase of compressed (nematic) aerogel. The orientation is reversed due to the opposite structure but the consistency of $\xi(T_x, P)$ remains.

characterize the path distribution of high porosity anisotropic aerogel. Furthermore, the mean free path in the anisotropic samples does not reveal the two length scales we find in both scattering data and simulation. The blue and orange curves in Fig. 5 b are quite different and yet they have the same first moment. We infer that it is insufficient for theoretical calculations of superfluid $^3$He in anisotropic aerogel to use the mean free path to represent the effects of anisotropy.

To recap, we find that the correlation function (see Methods), structure factor, and distribution of free paths form a set of metrics that can be used to characterize and classify anisotropic aerogels. From this process, it was determined that axially compressed silica aerogel has nematic strands while stretched aerogel has planar structure. We use this result in the following section to interpret superfluid $^3$He experiments that employ these aerogels.

## Anisotropic aerogel and superfluid $^3$He

Superfluid $^3$He is an unconventional, topological superfluid with quasiparticles forming $p$-wave (L = 1), spin-triplet (S = 1) Cooper pairs creating a manifold of possible phases. In the chiral A-phase, the Cooper pairs have a net orbital angular momentum, $\boldsymbol{\ell}_A$, with a vector order parameter. In the isotropic B-phase, the Cooper pairs exist in a superposition of all three components of spin and orbital angular momentum projections with total angular momentum $J = 0$. The relative stability of the phases is strongly affected by aerogel. In the pure superfluid, both the A and B-phases can exist as stable equilibrium phases depending upon temperature, pressure, and magnetic field.

This phase diagram is drastically altered in the presence of anisotropic aerogel. For compressed (nematic) aerogel in zero magnetic field, only the B-phase is observed for the entire pressure and temperature phase diagram[23,43]. For stretched (planar) aerogel, on the other hand, the A-phase becomes the equilibrium phase at all magnetic fields, pressures, and temperatures[8,44,45], in the absence of $^4$He pre-plating. Enhanced stability of the A-phase is consistent with both theoretical expectations[15,46] and other experiments notably in thin slabs[47]. Confirmation that stretched aerogel is indeed planar comes from recent observations of the magnetic susceptibility of surface Andreev bound states (SABS)[48]. The B-phase can be recovered in stretched

aerogels by pre-plating the surface of aerogel with $^4$He. The magnetic susceptibility in the B-phase is greatly enhanced by SABS, theoretically predicated for a planar-confined B-phase.

In addition to altering the stability of phases, anisotropic aerogel has been observed to reorient the orbital degrees of freedom as seen in Fig. 1a, b. In the presence of symmetry-breaking effects such as magnetic fields, boundaries, or anisotropic disorder, the B-phase orbital degrees of freedom are distorted giving rise to a preferred axis denoted $\boldsymbol{\ell}_B$. Recently, sharp transitions have been observed where the orbital vectors in the two phases spontaneously reorient by 90° uniformly across the entire system as temperature or pressure is varied[23,45]. It was determined that this reorientation is dependent upon the anisotropy of the aerogel and not from competing orienting effects such as from boundaries as has been reported previously in isotropic aerogel[49].

Phase identification of the superfluid, and identification of the direction of the angular momentum axis can be determined from NMR spectra obtained in a high homogeneity, steady magnetic field, discussed most recently by Zimmerman et al.[50] In the superfluid A-phase of stretched aerogel, $\boldsymbol{\ell}_A$ orients parallel to the anisotropy axis $\boldsymbol{\epsilon}$ at high temperature near the superfluid transition, $T_c$. NMR experiments[8,44,45] show that at a lower temperature, denoted $T_x$, $\boldsymbol{\ell}_A$ spontaneously flops to being perpendicular to $\boldsymbol{\epsilon}$ across the entire sample, as depicted in Fig. 6a. In the superfluid B-phase of compressed aerogel, $\boldsymbol{\ell}_B$ is initially perpendicular to $\boldsymbol{\epsilon}$ near $T_c$, the opposite of what is observed in the A-phase of stretched aerogel. At $T_x$, $\boldsymbol{\ell}_B$ sharply reorients to being parallel to $\boldsymbol{\epsilon}$ with a narrow transition width of ~15 $\mu K$[23]. This orbital-flop transition varies with pressure between ~0.67 $T_c$ at 7.5 bar to 0.88 $T_c$ at 26 bar. The opposite behavior of these two samples is resolved by considering the underlying structure of the aerogel comparing planar and nematic as we discuss next.

From both the SAXS data and $S(\boldsymbol{q})$, it is evident that these anisotropic aerogels have different structure at long and short length scales. The large-scale structure is given by the dipolar pattern while the small-scale structure is given by the ellipsoidal pattern at large $q$, Fig. 4 (also see Supplementary Materials). Furthermore, the scattering patterns reveal that large-scale structure is oriented perpendicular to the

small-scale structure. We propose that this structural crossover in the aerogel induces the orbital-flop transition. The most important length scale in a superfluid is the coherence length, $\xi$, which can be thought of as the size of a Cooper pair (or more accurately, the healing length for variations of the order parameter). The coherence length is largest near the superfluid transition and decreases with temperature. Therefore, at high temperature near $T_c$, the superfluid's orbital degrees of freedom will be sensitive to large-scale disorder. As $\xi$ becomes smaller at lower temperature, the smaller scale structure in the aerogel dominates.

We have shown that at long-length scales stretched aerogel has planar structure while compressed aerogel has a nematic structure. For planar aerogels, the surface normal of the large scale structure is parallel to the anisotropy axis. Correspondingly, for the A-phase of superfluid $^3$He in planar aerogel we would expect $_A \parallel \hat{\boldsymbol{e}}$ at high temperatures above $T_x$, and $_A\hat{\boldsymbol{e}}$ below[8,44,45]. If the preferred orientation of $\boldsymbol{\ell}$ is determined solely by aerogel structure it must be independent of the superfluid phase. Consequently, for a nematic aerogel, parallel and perpendicular orbital orientations are just interchanged. This behavior is consistent with experiment[23,44] and summarized in Reference[51].

The Ginzburg-Landau coherence length varies with both temperature and pressure given by[52]

$$\xi_{GL}(T,P) = \left[\frac{7\zeta(3)}{12}\right]^{1/2} \frac{\hbar\, v_F(P)}{2\pi\, k_B T_c(P)} \left(1 - T/T_c\right)^{-1/2}, \qquad (4)$$

where $\zeta$ is the Riemann-zeta function, $v_F(P)$ is the pressure-dependent Fermi velocity, and $T_c(P)$ is the pressure dependent superfluid transition temperature. The coherence length diverges near the second order phase transition and decreases with temperature as $(1 - T/T_c)^{-1/2}$. The temperature-independent prefactor $\xi_0(P)$ (not to be confused with the zero-temperature BCS coherence length $\xi_{BCS}$) varies from 15 to 80 nm between solidification pressure (34.4 bar) to 0 bar. The experiments in ref. 23 occur between 7.5 bar and 27 bar where $\xi_0(P)$ ranges from 34 to 18 nm. The GL coherence length is shown in Fig. 6 for various pressures.

Remarkably, $\xi(T_x/T_c)$ is essentially pressure independent, shown in Fig. 6. Furthermore, the fact that $\xi(T_x/T_c)$ ~ 50 nm, is consistent with a simple mechanism for the orbital-flop transition. Above $T_x$, $\xi$ is large so $\boldsymbol{\ell}$ is oriented perpendicular to the large-scale aerogel structure. At the temperature where $\xi$ drops below ~ 50 nm, the orbital flop transition occurs and $\boldsymbol{\ell}$ is reoriented by the small-scale structure. The crossover in scales seen in the correlation function and structure factor are of order ~ 20–50 $r_0$. For silica aerogel with $r_0$ ~ 1.5 nm, this corresponds to 30-75 nm compared with $\xi_{GL}(T_x, P)$ ~ 50 nm. The crossover is also evident in the SAXS data (see Supplementary Materials). This model suggests that at low pressure where the coherence length is substantially larger we expect $T_x$ to decrease as pressure is lowered. Below 1.5 bar, the coherence length does not drop below 50 nm and no crossover transition would be expected.

Other experiments using a different type of planar aerogel also observe a phase diagram dominated by the A-phase with the orbital angular momentum orienting perpendicular to planar sheets[33]. However, an orbital flop transition was not observed in those experiments because the aerogel has much stronger anisotropy and does not appear to have two different length scales.

The sharpness of the orbital flop transition creates a useful experimental tuning parameter for probing new physics since the angular momentum axis is the chiral axis in the A-phase. Recently, it was shown that there is a substantial anomalous thermal Hall effect in superfluid $^3$He in the presence of impurities like aerogel[53]. The direction of transverse thermal current is strongly dependent upon the orientation of the orbital angular momentum. Therefore, the orbital-flop can be used as a switch to turn on, or off, the transverse thermal gradient. Because of the sharpness of the transition, the Hall effect

should drop to zero abruptly as temperature is changed across $T_x$. This switching would be a definitive signature of the anomalous thermal Hall effect, an unambiguous indication of broken-time reversal symmetry.

## Summary

We have described a procedure to simulate and characterize anisotropic aerogels with planar and nematic strands. The anisotropy is induced by biasing the diffusion process and can be characterized by the autocorrelation function, structure factor, and distribution of free paths. We make a connection to experimental aerogel by comparing the shape of the SAXS pattern with the simulated structure factor. Both the calculated structure factor and the SAXS data exhibit a congruent dipolar shape at small-$q$ and a perpendicular ellipsoidal pattern at large-$q$. These two patterns reveal the two different length scales of anisotropy in the aerogels. From this connection, we are able to classify silica aerogel and show that stretched silica aerogel has large-scale planar structure while compressed aerogel has large-scale nematic structure. Finally, we provide a description of the aerogel's effect on the orbital angular momentum of superfluid $^3$He. The orbital angular momentum is oriented by the large scale structure in the aerogel at high temperature before spontaneously reorienting at a lower temperature due to the small-scale structure. Temperature-dependent sensitivity to the orientation of the order parameter of the superfluid is directly linked to the temperature dependence of the superfluid coherence length. We suggest that the "orbital-flop" transition can be leveraged in future work to observe the anomalous thermal Hall effect in superfluid $^3$He.

## Methods

### Simulation of anisotropic aerogel

Aerogel can be accurately simulated using the procedure detailed in section **II** of ref. 20. A random point field of $N$ particles (ranging from N = 5000 to 200,000) is initialized in a periodic box with volume $L^3$. The particles have a distribution of radii given by a log-normal distribution with a sample mean of $r_0$ and sample standard deviation $\sigma_0$. The mean radius $r_0$ sets the scale for comparing the simulated aerogel with real aerogel, therefore all lengths are given in units of $r_0$. Our work is focused on high porosity (low density) aerogel with the filling fraction of $\rho_0$ ~ 2%, where $\rho_0 = \frac{4}{3}\pi r_0^3 \frac{N}{L^3}$, corresponding to real silica aerogel with mass density ~45 mg/cm$^3$[32]. For $N = 200000$, the box size $L$ is 350 $r_0$ which is the figure of merit for determining finite size effects discussed later regarding the correlation function. The standard deviation is fixed at $\sigma_0 = r_0/8$ because it does not affect the large scale structure for reasonable values of $\sigma_0$. For each simulation step, a randomly chosen particle (or cluster of particles) is moved along a randomly chosen direction ($\pm\boldsymbol{x}$, $\pm\boldsymbol{y}$, or $\pm\boldsymbol{z}$ with equal probability) by a step size $\leq r_0/4$, depending upon the choice of $\epsilon$. For $\epsilon = 1$, all directions have the same step size of $r_0/4$. For $\epsilon < 1$, the step size along $\boldsymbol{z}$ is reduced to $(\epsilon r_0)/4$. For $\epsilon > 1$, the step sizes along $\boldsymbol{x}$ and $\boldsymbol{y}$ are reduced to $r_0/(4\epsilon)$.

The particles are allowed to diffuse randomly until they collide with another particle or cluster. If a collision occurs, the two particles are joined into a cluster and thereafter diffuse together. The diffusion coefficient is controlled by the mass, $m$, of the aggregate, with larger clusters diffusing more slowly. The probability that a cluster is moved at each time step is proportional to $m^{-\alpha}$, with $\alpha$ chosen to be between $0.5 - 1$ which results in aerogels with a fractal dimension in the range $1.7 - 1.8$[20,54]. When all particles are joined into a single cluster, the simulation ends yielding a gel.

The characteristic size of the planar and nematic structures can be extracted from the density variation along each axis. This is calculated by taking a planar cross section of aerogel with a thickness of 2 $r_0$. As this slice is translated in the direction normal to the plane, the total number of aerogel particles in the slice will vary. For planar aerogel, the

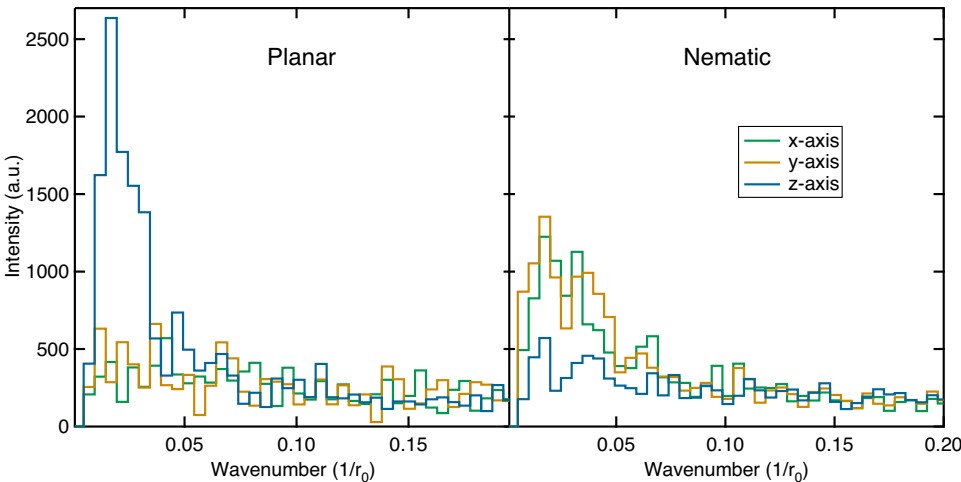

**Fig. 7 | Fourier transform of the density along each direction.** For planar aerogels $\epsilon = 8$ (left), there is a sharp peak near wave number $q = 0.015\,r_0^{-1}$ corresponding to a typical spacing between local maxima in density of about $60 - 70\,r_0$. The size of the gaps between planes is then roughly half of that, ~ $30\,r_0$. There is not much density variation in the $xy$-plane. For nematic aerogel $\epsilon = 0.125$ (right), there are peaks in the $x$- and $y$-axis density which can be interpreted as the diameter of the nematic bundles.

density variation along the z-axis will have a periodicity given by the thickness of the planar sheets and gaps, as seen in the left panel of Fig. 7, which is the Fourier transform of the density. For nematic aerogel, the opposite behavior is observed with density variations along the x- and y- axes. We find a consistent description of these nematic and planar structures from calculation of the autocorrelation function, the structure factor, and the distribution of geometric free paths presented in this work.

## Correlation functions

Silica spheres aggregate to form strands which cluster together to form larger structures that make up the aerogel network. The positions of the spheres are non-uniform and highly correlated in space. This non-uniformity is encoded in the autocorrelation function, $g$, which is the two-point characteristic of $\rho$ given by point-wise multiplication of $\rho$ evaluated for all pairs of particles. In the most general case, it is a function of all particle coordinates. This is reduced when certain assumptions and symmetry constraints are applied. For a globally homogeneous cluster of $N$ particles in a volume $V$, $g$ only depends upon the separation vector, $\boldsymbol{R}$, between two points given by

$$g(\boldsymbol{R}) = \frac{\langle \rho(\boldsymbol{r_i} + \boldsymbol{R})\, \rho(\boldsymbol{r_i}) \rangle}{N(N-1)/(2V)} \quad (5)$$

where the angled-brackets, $\langle \ldots \rangle$, represent an ensemble average over all pairs (the homogeneous assumption) and the denominator $N(N-1)/(2V)$ is the mean density of pairs. When normalized to the density of pairs, the autocorrelation function gives the excess likelihood to find two particles separated by a vector $\boldsymbol{R}$, relative to a random uniform Poisson point field of the same density[20,55,56]. The correlation function defined in Eq. (5) is also sometimes called the "pair correlation function", "pair distribution function", or "radial distribution function" depending upon the application[28,56,57]. It is useful to define $g$ in spherical coordinates, $g(R, \theta, \phi)$, where $\theta$ is the polar angle with respect to $\boldsymbol{z}$ and $\phi$ is the azimuthal angle in the $xy$-plane. In the limit of large separation, $R \to \infty$, $g(\boldsymbol{R}) \to 1$, indicating no excess correlation above that of a uniform distribution.

While different samples drawn from the same probability distribution (i.e. simulated with the same parameters or experimentally grown under the same conditions) will have a different value for the aerogel field $\rho(\boldsymbol{r})$ at any point $\boldsymbol{r_i}$, the two samples will have the same two-point functions because the correlations remain the same.

Therefore, $g(\boldsymbol{R})$ can be averaged between samples while $\rho$ cannot. In addition, most applications of aerogel do not depend on the location of the silica spheres themselves; but rather the open space between them, i.e. the complement of the aerogel structure. Characteristic cluster and void sizes can be determined from the correlation function. Excess correlation ($g > 1$) indicates clustering at that separation distance and direction while a deficit in correlation ($g < 1$) indicates voids.

For isotropic aerogels ($\epsilon = 1$), $g(\boldsymbol{R})$ further simplifies to $g(R)$ by averaging over the angular variables, $g(R) = \frac{1}{4\pi} \int g(\boldsymbol{R}) \sin\theta\, d\theta\, d\phi$. On the other hand, for anisotropic samples with $\epsilon \neq 1$, the angular-dependence of $g(\boldsymbol{R})$ becomes important. In the case of azimuthally symmetric, uniaxial anisotropy, $g$ is simplified by averaging only over the azimuthal angle yielding $g(R, \theta) = \frac{1}{2\pi} \int g(\boldsymbol{R})\, d\phi$. The correlation functions determined here have more structure than the correlation functions proposed in the literature which are simple power-laws with an upper fractal exponential cutoff[28]. This is not surprising as the aerogel is anisotropic with different macroscopic structure than for isotropic aerogels. As seen in Fig. 8, $g(R, \theta)$ for nematic and planar aerogel have non-uniform $\theta$-dependence, a clear signature of anisotropy. The $R$-dependence of the deficit in correlation gives the characteristic size of voids, while the $\theta$-dependence of the deficit gives the shape of voids for each sample. In all directions, there is a significant nearest-neighbor peak around $2\,r_0$ indicating contact between particles. The relative height of the nearest-neighbor peaks in different directions indicate whether pairing along $\boldsymbol{\epsilon}$ ($\cos(\theta) = \pm 1$) or in the plane perpendicular ($\cos(\theta) = 0$) is more likely. For nematic samples (the green curves in **c** and **d**), there is greater likelihood for the nearest neighbor to be in the plane perpendicular to $\boldsymbol{\epsilon}$ than parallel to $\boldsymbol{\epsilon}$ as seen in Fig. 8a. However, at intermediate separation, $10\,r_0 < R < 50\,r_0$, the direction of excess correlation swaps, indicating particles are more likely to be collimated along $\boldsymbol{\epsilon}$. This is the signature of long nematic strands.

The opposite behavior is observed for planar ($\epsilon > 1$) samples (the blue curves in panel Fig. 8 c, d). At small separations, there is preferential pairing along $\boldsymbol{\epsilon}$ but for larger separations, a neighbor is more likely to be found in the plane. Increasing $\epsilon$ increases the nearest-neighbor peak at short separations but also increases the deficit in correlation in the intermediate range of ~ $20\,r_0$. This can be interpreted to be the scale of the thickness of the planes of aerogel strands. A silica sphere located in the planes is less likely to have a neighbor above or below it at distances greater than the sheet thickness, but less than two

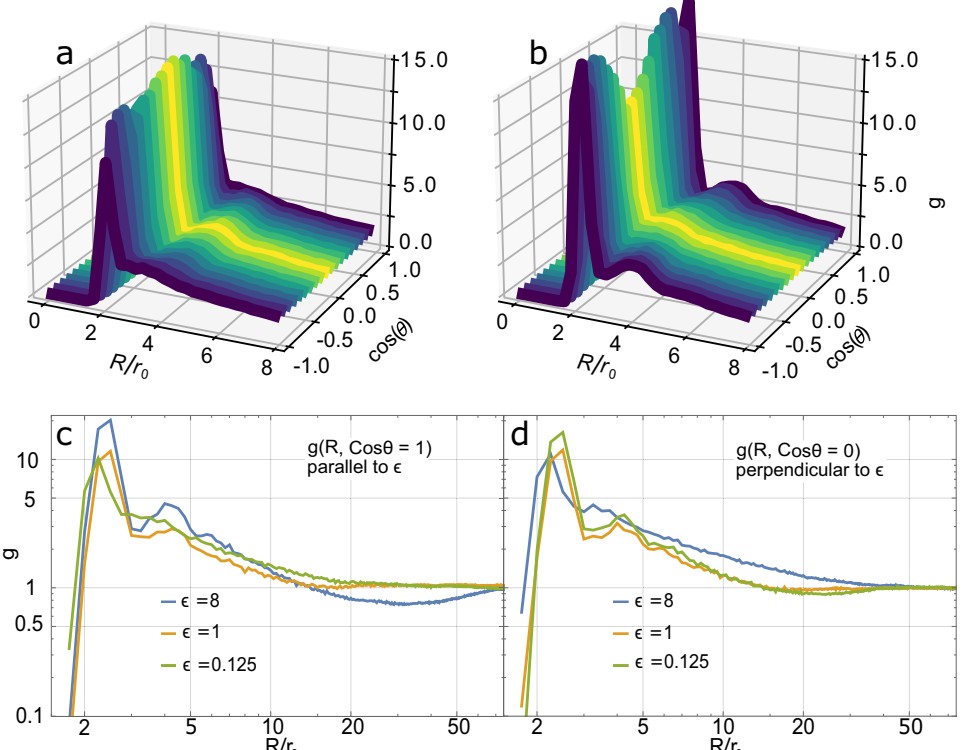

**Fig. 8 | Anistropic pair correlation of nematic and planar aerogels as a function of separation $R/r_0$ and cos($\theta$). a** For nematic ($\epsilon = 0.25$) and (**b**) for planar ($\epsilon = 4$), where cos($\theta$) = $\pm 1$ indicates pair correlations parallel to $\epsilon$ and cos($\theta$) = 0 indicates correlations in the perpendicular plane. The degree of excess and deficit in correlation can be tuned by changing $\epsilon$ as seen in the bottom two panels. **c** Pair correlations parallel to $\epsilon$ for various values of $\epsilon$. Increasing $\epsilon$ increases the nearest-neighbor peak at short separations but also increases the deficit in correlation in the intermediate range of $10 < R/r_0 < 50$. **d** Correlation perpendicular to $\epsilon$. Increasing $\epsilon$ in this direction has the opposite effect. The deficit in correlation in (**d**) is less than in **c**. This implies that the spacing between nematic strands for $\epsilon < 1$ is less than the gaps between sheets in the planar samples. In all cases, $g \to 1$ above ~ $70 r_0$ indicating no excess correlation above this scale.

sheet thicknesses. Correspondingly, as $\epsilon$ increases, so does the size of gaps between the sheets for planar aerogel. In both the $\epsilon > 1$ and $\epsilon < 1$ cases, it is the behavior of the correlation function at the intermediate length scale of 10 to 50 $r_0$ that is central to understanding the macroscopic properties of the aerogel. The correlation function is dominated by the nearest-neighbor peak at 2 $r_0$; but this only describes the small scale correlations.

We can also determine whether finite size effects are relevant for understanding these anisotropic aerogels. As seen in Fig. 8, the correlation approaches 1 for separations beyond 70 $r_0$ for both the nematic and planar aerogels indicating no complex structures above that scale. This is well below the typical box size used, L = 350 $r_0$. We conclude that the anisotropic properties of these aerogels are not limited by the box size.

There are two methods of numerically calculating g, the "direct" method and the Fourier-correlation method, each with their own advantages and disadvantages. The direct method simply applies the definition of the correlation function considering every pair of particles, calculates their separation vector $\boldsymbol{R}$, and histograms the set of $\boldsymbol{R}$ into equal volume R and $\theta$ bins. The raw bin counts are then normalized by the density of pairs and the bin volume, $\frac{N(N-1)}{2V} 2\pi r^2 dR\, d(\cos\theta)$. The radial bin width is $dR$ and the theta bin width is $d(\cos\theta)$. This method can be implemented natively in spherical coordinates but is slow in time, scaling as $\mathcal{O}(N^2)$ for N particles.

Because of finite sample size, particles near the edge of the sample box will have an artificial deficit in neighbors leading to the tail of the distribution (large $\boldsymbol{R}$) being incorrect. This can be corrected by several different methods as described in ref. 56. Astronomers calculating auto-correlation functions of galaxies observe biases for large $\boldsymbol{R}$[58] and have devised various estimators to correct for this. The central idea is

to consider a randomly distributed sample of similar size and density. This artificial sample will have the same finite size effects as the simulation of interest. The autocorrelations of the simulation of interest and of the artificial sample along with the cross-correlation *between* the artificial sample and the simulation are combined to remove finite size effects. Different combinations of autocorrelations and cross-correlation have been suggested each with their own statistical bias[56]. We have implemented several of the most widely used estimators and compared them with a simple procedure of cross-correlating the original aerogel sample $\rho(\boldsymbol{r})$ with a copy of itself that has been spatially-shifted in all three directions by the simulation box size, $\rho(\boldsymbol{r} \pm L(\hat{\boldsymbol{x}}, \hat{\boldsymbol{y}}, \hat{\boldsymbol{z}}))$. This procedure is equivalent to applying the periodic boundary used in the simulation. We find that the latter method effectively corrects the tail of $g(\boldsymbol{R})$ consistent with the other estimators without the need to generate a test sample. The correlations in Fig. 8 were determined using the direct method with periodic boundary conditions.

The second method uses the Fourier-correlation (Wiener–Khinchin) theorem and the efficiency of fast-Fourier transforms (FFT) to speed up the process. This is best done in the cartesian representation of the aerogel field, $\rho_{ijk}$. The cartesian autocorrelation function, $g_{xyz}$, is the inverse Fourier transform given by

$$g_{xyz} = \frac{\langle \rho_{ijk}\, \rho_{i'j'k'} \rangle}{N(N-1)/(2V)} = \frac{\mathcal{F}^{-1}\left\{ |\hat{f}_{lmn}|^2 \right\}}{N(N-1)/(2V)} \qquad (6)$$

This calculates the circular autocorrelation of $\rho(\boldsymbol{r})$ which naturally enforces the periodic boundary used in the simulation so it does not need to be corrected for finite size effects. Due to the speed and efficiency of FFT algorithms, this method is significantly faster in time and

scales only as $\mathcal{O}(NLogN)$. However, this method is memory intensive as the matrix representation of $\rho$ grows as $L^3$ for $L$ lattice sites in each dimension. For a cubic lattice with $10^3$ sites per dimension, $\rho_{ijk}$, stored as a 32-bit float, there will be ~4 Gb of data. In computational complexity, the FFT-correlation method scales well in time but poorly in space (memory), while the direct method is the opposite, efficient in space but slow in time. Importantly, the FFT method also calculates the three-dimensional, cartesian structure factor, $S_{xyz}$.

## Data availability
Experimental data and simulation results are available from the corresponding author upon request. A sample nematic aerogel ($\epsilon = 0.5$) can be found at https://github.com/Halperin-Lab/DLCA.

## Code availability
Simulation and analysis code are available from the corresponding author upon request. Compiled DLCA program can be found at https://github.com/Halperin-Lab/DLCA, https://doi.org/10.5281/zenodo.10086252.

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

## Acknowledgements

The authors thank Thomas Haard whose original DLCA code formed the starting point for our simulation code. We also thank Jim Sauls, Johannes Pollanen, John Davis, and Jia Li for their advice and discussion. Support is acknowledged by the Northwestern University undergraduate Integrated Science Program (YCCT). This work was supported by the National Science Foundation, Division of Materials Research, grant DMR-2210112.

## Author contributions

Man Nguyen: simulations, superfluid $^3$He, and manuscript preparation; Joshua Simon: simulations; John Scott: superfluid $^3$He experiments and manuscript preparation; Andrew Zimmerman: simulations and superfluid $^3$He experiments; Yun-Chieh (Cincia) Tsai: aerogel preparation; William Halperin: superfluid $^3$He and manuscript preparation.

## Competing interests

The authors declare no competing interests.
