## [Peer Review File · Nature Communications]

Orbital-flop Transition in Superfluid ^3He in Anisotropic Silica AerogelREVIEWER COMMENTS

Reviewer #1 (Remarks to the Author):

Superfluid Helium 3 has a diverse phase diagram and the order parameter has an intricate group structure. The present paper studies He-3 immersed in anisotropic silica aerogel, and it is found that the sign of the applied strain selects either the chiral A-phase or the time-reversal-symmetric B-phase. The authors propose that this structure is responsible for the orbital analogue of the spin-flop textural transition in superfluid He-3.

Using different values of a parameter ϵ enables either nematic ($\epsilon < 1$) or planar ($\epsilon > 1$) aerogels to be created. At a certain temperature T_x , the angular momentum vector of the superfluid changes orientation, and based on this result it was determined that axially compressed silica aerogel has nematic strands whereas stretched aerogel has a planar structure. The experimental and theoretical analyses in the paper complement one another, and a thorough discussion concerning the structure-factor and Fourier-transform computations is provided. The modification of the usual He-3 phase diagram due to the presence of an aerogel is an innovative way of selecting either the A or B phases of He-3. Overall the paper is a valuable addition to the literature on He-3 and I am in favour of publication. I have several comments and queries below which would be good to clarify (in the text if relevant).

(i) Repeated "the" on line 66.

(ii) Lines 81-84 (and elsewhere); In keeping with the nomenclature for liquid crystals, the planar phase can be equivalently described as a "smectic" phase. It might be useful to mention this for the benefit of the reader, since smectic liquid crystals are quite prevalent in condensed matter physics – (Larkin-Ovchinnikov superfluids, the striped phase in confined He-3 etc.)

(iii) Figures 6 (d), (e), and (f) would be much more informative if the black square were not covering the image.

(iv) Line 257. Can the authors elaborate further on why their findings do not coincide with Volovik's arguments?

(v) Line 375. If the mean-free path cannot be used to represent the effects of anisotropy (as stated on line 329), then what is the length scale to compare with the coherence length?

(vi) On line 391, $\xi_0(P)$ is referred to as the zero-temperature coherence length. This is incorrect. In BCS theory, the zero-temperature coherence length is $\hbar v_F / [\pi \Delta(0)]$. The coherence length in line 391 is a temperature-dependent coherence length that is (usually) evaluated near T_c , where GL theory is applicable. Furthermore, in arbitrary D spatial dimensions, the denominator in ξ^2 is $16D$, and so, for $D=3$, the result should be 48, not 24 as presently written. [See the discussion around Eq. 6.16, Larkin and Varlamov "Theory of fluctuations in superconductors".] The authors should clarify the discussion concerning the coherence length and emend any related calculations and or figures if necessary.

(vii) There has been a lot of recent interest in studying confined He-3 in slab geometries, where possible striped phases have been discovered. Is there any possibility of using aerogels to see similar physics in He-3?

Reviewer #2 (Remarks to the Author):

The paper entitled "Orbital-flop Transition in Superfluid ^3He in Anisotropic Silica Aerogel" is well written, breaks new ground and presents novel and timely results. It is entirely appropriate for publication in Nature Communications.

It is mainly concerned with simulations on the structure factor of aerogels, with included anisotropy. These results are compared to measured structure factors and found to be in good agreement.

The authors take pains to also compare the observed spin-flop transition to the structure of the simulated aerogels and find good agreement.

This is all very useful and will serve as a benchmark for future studies - both experimental, theoretical and simulation. The really useful section is the results section with a very useful discussion of Levy flights in these materials.

I do have two suggestions.

1. The authors should state whether the inferred structure factor provides insight into the extreme metastability of the A phase vs the B phase. It is surprising to me that this is not mentioned. Perhaps it is because nothing is learnt from this study regarding that question - which is fair enough but I think its worth a paragraph.

2. The color scheme adopted in Fig 2c, 3 a, b is very difficult to read. Perhaps they could use some different (less "dark") color pairing.

Reviewer #3 (Remarks to the Author):

The paper deals with the orbital-flop transition of superfluid ^3He in anisotropic silica aerogels and presents an approach to classify aerogels as either "planar" or "nematic" through diffusion limited cluster aggregation simulations with an additional comparison to small angle X-ray measurements. The authors conclude that the orbital-flop is caused by competition between short- and long-range structures in these aerogels. The work grants insights into the impact of impurities and their anisotropy in superconductors. The proposed DLCA simulations are based on the work by Hasmy et al. and incorporate a modified procedure for biasing the diffusion process along a specific axis.

Strengths: The paper appears interesting from a modeling point of view and presents a novel application of the DLCA algorithm for quantifying anisotropic silica aerogels. It is well written and logically structured. In particular, it concerns the methodology section describing the calculation of the pair correlation function as well as the structure factor. The results obtained especially concerning anisotropy induced through biasing the diffusion process seem plausible due to the comparison to SAXS patterns.

Weaknesses: The paper is very vague on the implementation of the DLCA algorithm and only refers to a compiled program on GitHub. In the 30-year-old implementation of Hasmy et al. single aggregates are connected to each other in a hierarchical scheme. Was this done exactly in the same manner? Many other implementations such as e.g. by Heinson et. al [1], Zhu et. al [2], Jungblut et. al [3], Abdusalamov et. al [4] (see also reference therein) have been proposed since then but are not mentioned by the authors. In the implementation all length-related values are normalized to the mean radius r_0 , however this value is not specified. Additionally, from the point of view of the implementation it does not become apparent why the final and initial box sizes need to be provided. There are several parameters (such as e.g. the mean particle size, the standard deviation of the distribution of the particle radii, the three step size in each direction, the three offset for step directions) which can be changed. However, it is not explained how these parameters were obtained and whether they were correlated to experimental results. Furthermore, the effects of randomness of the random walk process is not elaborated.

In summary, I could recommend this paper for the publication only after a thorough revision according to the above comments. In particular, a more detailed explanation of the underlying algorithm and its functionality should be added at least in the supplementary information.

Additional references

[1] Heinson, W. R., et al. "Divine proportion shape invariance of diffusion limited cluster-cluster aggregates." *Aerosol Science and Technology* 49.9 (2015): 786-792.

[2] Zhu, Chuan-Yong, et al. "Numerical modeling of the gas-contributed thermal conductivity of aerogels." *International Journal of Heat and Mass Transfer* 131 (2019): 217-225.

[3] Jungblut, S., et al. "Diffusion-and reaction-limited cluster aggregation revisited." *Physical Chemistry Chemical Physics* 21.10 (2019): 5723-5729.

[4] Abdusalamov, R., et al. "Modeling and simulation of the aggregation and the structural and mechanical properties of silica aerogels." *The Journal of Physical Chemistry B* 125.7 (2021): 1944-1950.

Response to Reviewers

Reviewer 1:

- i) Repeated “the” on line 66

Fixed grammatical error on line 66.

- ii) Lines 81-84 (and elsewhere); In keeping with the nomenclature for liquid crystals, the planar phase can be equivalently described as a “smectic” phase. It might be useful to mention this for the benefit of the reader, since smectic liquid crystals are quite prevalent in condensed matter physics.

We have added this suggestion after introducing planar aerogel at line 94.

- iii) Figures 6 (d), (e), and (f) would be much more informative if the black square were not covering the image.

The black square cannot be removed since it is the shadow of the beam stopper for the X-ray detector, there is no data in that region. We have made this fact more explicit in the caption.

- iv) Line 257. Can the authors elaborate further on why their findings do not coincide with Volovik's arguments?

Volovik argues (JLTP **150** 453-463 (2008), Fig. 3) that stretching creates nematic aerogel and while compressing creates planar aerogel. He is correct that the angular momentum should orient perpendicular to the large scale structure but incorrectly predicts which type of strain produces nematic and planar aerogel.

- v) Line 375. If the mean-free path cannot be used to represent the effects of anisotropy (as stated on line 329), then what is the length scale to compare with the coherence length?

The length scale used to compare with the coherence length is a “crossover length” that separates short and long length scales apparent in both the SAXS data and structure factor calculations (the crossover from dipolar to ellipsoidal patterns in Fig. 6 and Fig. 9) and also calculated correlation functions. We have added a figure in supplementary materials showing this explicitly, Fig. 10.

- vi) On line 391, $\xi_0(P)$ is referred to as the zero-temperature coherence length. This is incorrect. In BCS theory, the zero-temperature coherence length is $\hbar v_F / [\pi \Delta(0)]$. The coherence length in line 391 is a temperature-dependent coherence length that is (usually) evaluated near T_c , where GL theory is applicable. Furthermore, in arbitrary D spatial dimensions, the denominator in ξ^2 is $16D$, and so, for $D=3$, the result should be 48, not 24 as presently written. [See the discussion around Eq. 6.16, Larkin and Varlamov “Theory of fluctuations in superconductors”.] The authors should

clarify the discussion concerning the coherence length and emend any related calculations and or figures if necessary.

Our definition of the coherence length is taken from *The Superfluid Phases of Helium 3*, Vollhardt and Wölfle (equation 7.18b and 7.18c, page 195).

- vii) There has been a lot of recent interest in studying confined He-3 in slab geometries, where possible striped phases have been discovered. Is there any possibility of using aerogels to see similar physics in He-3?

We have yet to observe a striped phase in this system. This is a very interesting suggestion which we plan to pursue in future work.

Reviewer #2:

- 1) The authors should state whether the inferred structure factor provides insight into the extreme metastability of the A phase vs the B phase. It is surprising to me that this is not mentioned. Perhaps it is because nothing is learnt from this study regarding that question - which is fair enough but I think its worth a paragraph.

We added commentary about stability of the A and B phases in these aerogels where planar aerogels generally prefer the A-phase while nematic aerogels prefer the B-phase (starting at line 353).

- 2) The color scheme adopted in Fig 2c, 3 a, b is very difficult to read. Perhaps they could use some different (less "dark") color pairing.

Thank you for the suggestion. We have tried several other brighter colormaps, but the current option gives the most contrast.

Reviewer #3:

Weaknesses: The paper is very vague [sic] on the implementation of the DLCA algorithm and only refers to a compiled program on GitHub. In the 30-year-old implementation of Hasmy et al. single aggregates are connected to each other in a hierarchical scheme. Was this done exactly in the same manner?

The basis of our simulation code follows Hasmy *et al.*, Phys. Rev. B. **50**, 6006 (1994) using what they called “DLCA in a box”, not the hierarchical scheme used in the very similar reference, Hasmy *et al.*, Phys. Rev. B. **50** 1305(R) (1994). The pseudo-code we outlined in our “Methods” section is an accurate and concise summary of this procedure. We have included more details in this summary.

Many other implementations such as e.g. by Heinson et. al [1], Zhu et. al [2], Jungblut et. al [3], Abdusalamov et. al [4] (see also reference therein) have been proposed since then but are not mentioned by the authors.

We have added additional references.

[25] R. Abdusalamov, C. Scherdel, M. Itskov, B. Milow, G. Reichenauer, and A. Rege, The Journal of Physical Chemistry B 125, 1944 (2021).

[28] W. Heinson, C. Sorensen, and A. Chakrabarti, Journal of Colloid and Interface Science 375, 65 (2012).

[35] C. Oh and C. M. Sorensen, Phys. Rev. E 57, 784 (1998).

In the implementation all length-related values are normalized to the mean radius r_0 , however this value is not specified.

For real silica aerogels used in superfluid ^3He experiments, r_0 is $\sim 1.5\text{--}2$ nm. We have specified r_0 in the text.

Additionally, from the point of view of the implementation it does not become apparent why the final and initial box sizes need to be provided.

We included the ability to change box size in the code to see if anisotropic aerogel could be generated by stretching the box after initializing an isotropic sample. This did not lead to anything interesting. It remains in the simulation code but is not used in the current work. This is explained in the Github readme.

There are several parameters (such as e.g. the mean particle size, the standard deviation of the distribution of the particle radii, the three step size in each direction, the three offset for step directions) which can be changed. However, it is not explained how these parameters were obtained and whether they were correlated to experimental results.

We have included relevant simulation parameters in the Methods section. As for other simulation parameters (such as density, variance in particle size), the variation in those parameters do not lead to anisotropy, which is the focus of this work. The choice of 2% density is consistent with aerogels used in experiment.

Furthermore, the effects of randomness of the random walk process is not elaborated.

We have given further details about how the random walk process is implemented in the Methods section.

REVIEWER COMMENTS

Reviewer #1 (Remarks to the Author):

The authors have satisfactorily addressed all but one of my comments. The discussion concerning the coherence length on page 5 is still unsatisfactory. The authors referred to Vollhardt and Wölfle (equation 7.18b and 7.18c, page 195); however, the discussion in that chapter is centred around the GL expansion near T_c ; thus, ξ_0 as defined is not the true zero-temperature coherence length. For example, in Eqs. (1.16-1.17) of Kopnin "Theory of Non equilibrium superconductivity", Kopnin refers to the expression for ξ_0 as the "zero-temperature" (with quotations) coherence length; Larkin and Varlamov are also careful to clarify that ξ_0 is not the same thing as ξ_0^{BCS} . The authors should clarify (in a footnote) on line 403 where the ξ_0 expression comes from and how it differs from ξ_0^{BCS} .

Reviewer #2 (Remarks to the Author):

The revised manuscript fully resolves any issues that I had noted in my first review. It meets all the criteria for publication which should move forward.

Reviewer #4 (Remarks to the Author):

I have been asked to focus this review on the model of aggregation presented. Therefore, this review will focus on the Methods section. Unfortunately, I have many concerns about this section. It is possible that they can all be chalked up to poor explanations, but it is also possible that the methods used are deficient. Given the lack of detail, I cannot tell, but at present I am strongly against publication.

First, the presentation of the numerical methods mixes dimensional and dimensionless quantities in a confusing manner. The authors seem to use the same notation to sometimes have dimensional units and sometimes be non-dimensionalized. They refer to a variance σ_0^2 and then to a variance σ_0/r_0 (The latter is probably a dimensionless standard deviation rather than a variance). They also mention a box size, L , which may be made dimensionless using r_0 , but in any case no value of the box size is provided and no discussion of its effects are given. Presumably only structures smaller than L can be reliably observed, but that is not discussed. In fact, the simulations are run until a single cluster is formed, which guarantees that finite-size effects will come into play. The authors mention using the particle radius r_0 as a typical length scale to which they compare other length scales, but then they go on to set r_0 to 4 simulation units. If r_0 is the typical length scale, then the dimensionless radius used should be 1. If instead every length is compared to r_0 , then its value is unimportant but giving its value in terms of "simulations units" is confusing. What is a simulation unit and why is it not r_0 ?

The presentation continues to be confusing due to bad notation use. The definition of the density, ρ , as a function of a position vector r is unclear. The authors state that for a discrete field of spheres, $\rho(r)$ is a list radii and center position of sphere. However, the authors go on to use ρ as a continuous function (not a list) defined everywhere and they never explain how they obtain it. Similarly, the autocorrelation function introduced, g , use varying notation. It is at first a function of all sphere centers, then of a single vector connecting any two sphere centers (denoted either as R_{ij} or R depending on the line), or a function of the length and angle to the vertical of that

vector, all with the same notation. These can all be fixed, but at the moment this is poorly explained to the point is being unclear as to what was actually computed.

The most important issue I have, however, concerns the presentation of the results which is either unclear or outright misleading. In Figure 2, the authors use a setup that I have only ever seen used to indicate that a portion of a figure was magnified to show more detail. However, what should be a magnification of an experimental picture is in fact a simulation result. This makes it seem like the simulations match the experiments very well, but in fact they are not compared directly at all. I am not certain because the text is not so clear, but I believe a similar misleading technique is used in figure 3 where the projection of a simulation obtained from spheres is "magnified" and where suddenly cylinders are shown. In fact, my biggest concern is that in their explanation of the model, the authors only ever mention that spheres or various sizes are moved randomly with steps taken along a grid but sometimes with a preferred direction. However, in the discussion and in Figure 3 strands (or cylinders) are shown. If strands were simulated, the authors never explain it. Moreover, if strands are to be simulated, their Brownian rotation should be considered, and that is never mentioned either. In fact, cluster rotation may also need to be included as well, see the work of Jungblut et al. and Polimeno et al., or at least an argument should be given for why it is neglected.

Because of these issues, and most in particular because the nematic aspect of the clusters seems to be of prime importance but its simulation is never explained, I do not favor publication at this point.

Reviewer 1:

The authors have satisfactorily addressed all but one of my comments. The discussion concerning the coherence length on page 5 is still unsatisfactory. The authors referred to Vollhardt and Wölfle (equation 7.18b and 7.18c, page 195); however, the discussion in that chapter is centred around the GL expansion near T_c ; thus, ξ_0 as defined is not the true zero-temperature coherence length. For example, in Eqs. (1.16-1.17) of Kopnin “Theory of Non equilibrium superconductivity”, Kopnin refers to the expression for ξ_0 as the “zero-temperature” (with quotations) coherence length; Larkin and Varlamov are also careful to clarify that ξ_0 is not the same thing as ξ_{BCS} . The authors should clarify (in a footnote) on line 403 where the ξ_0 expression comes from and how it differs from ξ_{BCS} .

We have removed the term “zero-temperature coherence length” to avoid confusion and added a comment noting the distinction between ξ_{GL} and ξ_{BCS} on page 21.

Reviewer 2:

The revised manuscript fully resolves any issues that I had noted in my first review. It meets all the criteria for publication which should move forward.

Reviewer 4:

We thank Reviewer 4 for corrections and helpful comments and have modified our manuscript accordingly. The specific points are made in sequence below. However, we emphasize here an important achievement of the simulation relevant to the referee’s points 6. and 7. A direct comparison of the structure factor of the simulated aerogels with small angle X-ray scattering (SAXS) of lab-grown aerogel is performed and shown to be in good agreement. This comparison is displayed in Fig. 6 along with associated text. Our result validates our anisotropic DLCA protocol and analytical interpretations of the simulated aerogels, clearly stated in both abstract and the introduction.

1. *First, the presentation of the numerical methods mixes dimensional and dimensionless quantities in a confusing manner. The authors seem to use the same notation to sometimes have dimensional units and sometimes be non-dimensionalized. They refer to a variance σ_0^2 and then to a variance σ_0/r_0 (The latter is probably a dimensionless standard deviation rather than a variance).*

Thank you for the correction, we have made changes beginning at line 56 (in red).

2. *They also mention a box size, L , which may be made dimensionless using r_0 , but in any case no value of the box size is provided and no discussion of its effects are given. Presumably only structures smaller than L can be reliably observed, but that is not discussed. In fact, the simulations are run until a single cluster is formed, which guarantees that finite-size effects will come into play.*

Further details have been added in the discussion of Fig. 5 (beginning on page 4 and continuing on page 10), including the fact that the box size, L , is $350 r_0$. This is sufficiently large that edge effects are insignificant since the correlation function approaches 1 for separations larger than $100 r_0$.

3. *The authors mention using the particle radius r_0 as a typical length scale to which they compare other length scales, but then they go on to set r_0 to 4 simulation units. If r_0 is the typical length scale, then the dimensionless radius used should be 1. If instead every length is compared to r_0 , then its value is unimportant but giving its value in terms of "simulations units" is confusing. What is a simulation unit and why is it not r_0 ?*

We emphasize the choice of r_0 does not affect the simulations and is only used to set the scale to compare with real aerogel. We have modified the Methods section to clarify this beginning at line 56.

4. *The presentation continues to be confusing due to bad notation use. The definition of the density, ρ , as a function of a position vector r is unclear. The authors state that for a discrete field of spheres, $\rho(r)$ is a list radii and center position of sphere. However, the authors go on to use ρ as a continuous function(not a list) defined everywhere and they never explain how they obtain it.*

We explicitly define the aerogel field $\rho(\mathbf{r})$ to be a function defined everywhere with compact support given by

$$\rho(\mathbf{r}) = \left\{ \begin{array}{ll} 1, & |\mathbf{r} - \mathbf{r}_i| \leq \varrho_i \\ 0, & \text{otherwise} \end{array} \right\}, \quad (1)$$

where \mathbf{r}_i and ϱ_i are the center and radius of the i^{th} -particle. This is stored numerically simply as a list of the \mathbf{r}_i and ϱ_i . There are no complications treating it as essentially continuous since derivatives of $\rho(\mathbf{r})$ are not required. The text has been modified as shown above.

5. *Similarly, the autocorrelation function introduced, g , use varying notation. It is at first a function of all sphere centers, then of a single vector connecting any two sphere centers (denoted either as $R_{i,j}$ or R depending on the line), or a function of the length and angle to the vertical of that vector, all with the same notation.*

The autocorrelation function has the same definition in all cases. It is only the number of relevant arguments that gets progressively reduced as various assumptions and symmetry constraints are imposed. We discuss its use, calculations, appropriate references to the literature, and specific statements relevant to our new protocol of a uniaxially biased DLCA simulation. We cannot find any instance of "varying notation" and consequently we respectfully disagree

with any implied criticism from the referee.

6. *In Figure 2, the authors use a setup that I have only ever seen used to indicate that a portion of a figure was magnified to show more detail. However, what should be a magnification of an experimental picture is in fact a simulation result. This makes it seem like the simulations match the experiments very well, but in fact they are not compared directly at all.*

As a helpful introduction to the structure of aerogels we show in Fig. 2 an image from scanning electron microscopy (SEM), now modified in the revised manuscript, to be a close-up view that is more instructive, placed adjacent to a portion of a simulated aerogel. A direct comparison of simulation and experiment is not made in this figure. This is neither inferred nor claimed; rather, as we have noted at the beginning of our response, it is quite adequately provided in Fig. 6.

7. *I am not certain because the text is not so clear, but I believe a similar misleading technique is used in figure 3 where the projection of a simulation obtained from spheres is "magnified" and where suddenly cylinders are shown. In fact, my biggest concern is that in their explanation of the model, the authors only ever mention that spheres or various sizes are moved randomly with steps taken along a grid but sometimes with a preferred direction. However, in the discussion and in Figure 3 strands (or cylinders) are shown. If strands were simulated, the authors never explain it. Moreover, if strands are to be simulated, their Brownian rotation should be considered, and that is never mentioned either. In fact, cluster rotation may also need to be included as well, see the work of Jungblut et al. and Polimeno et al., or at least an argument should be given for why it is neglected*

Cylinders or strands are not simulated. The only objects in the simulations are the spheres and their aggregates. The latter form an anisotropic gel. The process can be described as a coarse-grained hierarchy. At the most microscopic level, there are spheres; these form clusters that we call strands. At the highest level, they form planar and nematic structures arising from the anisotropic diffusion. We have clarified our terminology in the manuscript on page 5. Since there is excellent agreement between SAXS experiment and simulation, more complex protocols such as cluster rotation are not required and may not be relevant.

REVIEWERS' COMMENTS

Reviewer #1 (Remarks to the Author):

The authors have emended the manuscript, and it is suitable for publication.

Reviewer #4 (Remarks to the Author):

This resubmission has addressed several of the criticism I raised earlier. There remains some clarifications to make, but these should be easy to do.

Specifically, the authors clarified that some of the representations they have in Figure 3 are cartoons rather than scientific images, which is important to give the reader the correct perspective. They have also improved the explanations of some of the quantities they introduce.

However, the authors still have difficulty writing mathematically consistent expressions. At present, the mathematics are described with a "you know what I mean" approach rather than with the clarity and correctness that I believe is required for publication. As they state in their response, they do not see the issue, which explains why they cannot fix it. I will therefore make very specific recommendations for these small but important explanations, please see below.

In addition to the changes outline below, I also suggest, but do not insist, that the last paragraph on page 8 be relocated to a later section as it is more a discussion than a method description.

Improvement to make to mathematical notation and descriptions.

On line 65, please add: "with equal probability".

Equation 1: please add on the first line "if $\min_i |\vec{r} - \vec{r}_i| / \rho_i \leq 1$ for any $i=1... N$

Line 107: Please state precisely how thick are the planar cross-sections used.

Line 128: Do not use introduce the notation $g(r_i, \dots, r_j)$. Introduce $g(\vec{R})$ only.

Equation 2: write $g(\vec{R}) = \frac{\langle \rho(\vec{r}_j + \vec{R}) \rho(\vec{r}_j) \rangle}{N(N-1)/(2V)}$

so that the definition of the function directly involves the argument of the function.

Note that this way, the function g is well-defined as having a single vector argument, rather than having multiple conflicting definitions.

Line 137. Introduce the notation you will use for spherical coordinates: "We express \vec{R} in spherical coordinates as $\vec{R} = R (\cos \phi \sin \theta, \sin \phi \sin \theta, \cos \theta)$, where θ is the polar angle with respect to the anisotropy axis \vec{z} ."

Line 146. Write: "i.e. the complement of the aerogel structure"

Line 149. Do not use $g(\mathbf{R})$, since g is a function with a vector input. Instead, state that in that case $\frac{\partial g}{\partial \theta} = \frac{\partial g}{\partial \phi} = 0$.

Line 152. Say that $\frac{\partial g}{\partial \phi} = 0$, but $\frac{\partial g}{\partial \theta} \neq 0$.

Line 156. Use $g(\vec{\mathbf{R}})$

Line 187 and subsequent lines. The direct method without copies of the domain to implement periodicity is simply wrong. Please only give the description of the correct method that uses copies of the domain.

We have reordered the sections to comply with Nature Communications guidelines. The Methods section is now after the Results section. Certain paragraphs have been moved from Methods to Results to suit the new arrangement. We have highlighted those changes in blue. All other changes are in red.

Reviewer #1 :

The authors have amended the manuscript, and it is suitable for publication.

Reviewer #4 :

In addition to the changes outlined below, I also suggest, but do not insist, that the last paragraph on page 8 be relocated to a later section as it is more a discussion than a method description.

We have moved the paragraph to results section, now on page 6 (line 87).

1) "On line 65, please add: "with equal probability."

The text is modified accordingly, now at line 356.

2) "Equation 1: please add on the first line "if $\min_i |\vec{r}_i - \vec{r}_j| / \rho_i \leq 1$ for any $i=1 \dots N$ "

We have modified the first equation as recommended and moved it to the main text (line 52).

3) Line 107: Please state precisely how thick are the planar cross-sections used.

We have specified the thickness, now at line 368.

4) Line 128: Do not use introduce the notation $g(r_i, \dots, r_j)$. Introduce $g(\vec{R})$ only. Equation 2: write $g(\vec{R}) = \frac{\langle \rho(\vec{r}_j + \vec{R}) \rho(\vec{r}_j - \vec{R}) \rangle}{N(N-1)/(2V)}$ so that the definition of the function directly involves the argument of the function. Note that this way, the function g is well-defined as having a single vector argument, rather than having multiple conflicting definitions.

As recommended we have clearly defined the various usages of g . We note that it is standard notation to use the same g in all cases, even when the number of arguments gets reduced and simplified, cf. Ref. [57] McQuarrie, *Statistical Mechanics* section 13-2.

5) Line 137. Introduce the notation you will use for spherical coordinates: "We express \vec{R} in spherical coordinates as $\vec{R} = R (\cos \phi \sin \theta, \sin \phi \sin \theta, \cos \theta)$, where θ is the polar angle with respect to the anisotropy axis \vec{z} .

We have defined the coordinates as recommended, now at line 391.

6) Line 146. Write: "i.e. the complement of the aerogel structure"

The text is modified accordingly, now at line 401.

7) Line 149. Do not use $g(R)$, since g is a function with a vector input. Instead, state that in that case $\frac{\partial g}{\partial \theta} = \frac{\partial g}{\partial \phi} = 0$.

$g(R)$ has been defined explicitly, now at line 405.

8) Line 152. Say that $\frac{\partial g}{\partial \phi} = 0$, but $\frac{\partial g}{\partial \theta} \neq 0$.

$g(R, \theta)$ has been defined explicitly, now at line 408.

9) Line 156. Use $g(\vec{R})$

Same as 8).

10) Line 187 and subsequent lines. The direct method without copies of the domain to implement periodicity is simply wrong. Please only give the description of the correct method that uses copies of the domain.

In all figures and calculations used in the text, we apply the tail-correction algorithm. When we first introduce the numerical calculation of g , we immediately follow this definition with the discussion of the correct way to calculate the tail of the function. It is made clear that one cannot ignore finite size effects for the calculation.